# Transporter modulation of molnupiravir and its metabolite β-D-N4-hydroxycytidine across the blood-brain barrier in a rat

Chun-Hao Chang[1], Wen-Ya Peng [1], Wan-Hsin Lee[1], Ling Yang[1], Tung-Yi Lin[1], Muh-Hwa Yang [2] & Tung-Hu Tsai [1,3,4,5 ✉]

## Abstract

**Background** The antiviral drug molnupiravir is an orally bioavailable prodrug of the nucleoside analog β-D-N4-hydroxycytidine (NHC), which is used to treat coronavirus disease 2019 (COVID-19). However, there is very little information on the barrier distribution of molnupiravir. Our hypothesis is that molnupiravir and NHC can penetrate the blood–brain barrier (BBB) into brain tissue and that nucleoside transporters (equilibrative nucleoside transporters; ENT and concentrative nucleoside transporters; CNT) can modulate this process.

**Methods** To investigate the mechanism of molnupiravir transport through the BBB, multiple microdialyses coupled to a validated ultra-high-performance liquid chromatography tandem mass spectrometry (UHPLC–MS/MS) was developed to monitor dialysates, and nitro-benzylthioinosine (NBMPR; an inhibitor of ENT) was administered concomitantly with molnupiravir (100 mg/kg, i.v.) in the male rat.

**Results** Here, we show that molnupiravir is rapidly metabolized to NHC in the blood and crossed the BBB in 20 min. Furthermore, when NBMPR is concomitantly administered to inhibit efflux, the concentrations of molnupiravir and NHC in the brain increased significantly.

**Conclusions** In summary, molnupiravir rapidly transforms into NHC and crosses the BBB and reaches the brain at approximately 0.3-0.8% of the blood–brain ratio. The maximum concentration of NHC in the blood and brain is above the average half maximal inhibitory concentration (IC50) of the drug required to treat severe acute respiratory syndrome coronavirus 2 (SARS-CoV-2) infection, suggesting a therapeutic effect. The penetration of NHC is modulated by NBMPR. These findings provide constructive information on brain disorders in clinical patients with COVID-19.

## Plain language summary

Due to the global pandemic of coronavirus disease 2019 (COVID-19) caused by severe acute respiratory syndrome (SARS-CoV-2), molnupiravir is used orally to treat COVID-19 with emergency use authorization. However, it is not well understood whether molnupiravir and its active component can cross the blood–brain barrier. The aim of the study was to develop an experimental mouse model to monitor the journey of molnupiravir and its active component through the bloodstream and eventually into the brain. Our experimental data suggest that a therapeutically useful amount of molnupiravir crosses from the bloodstream into the brain.

[1] Institute of Traditional Medicine, School of Medicine, National Yang Ming Chiao Tung University, Taipei 112, Taiwan. [2] Institute of Clinical Medicine, National Yang Ming Chiao Tung University, Taipei, Taiwan. [3] Graduate Institute of Acupuncture Science, China Medical University, Taichung 404, Taiwan. [4] Department of Chemistry, National Sun Yat-Sen University, Kaohsiung 804, Taiwan. [5] School of Pharmacy, Kaohsiung Medical University, Kaohsiung 807, Taiwan. ✉email: thtsai@nycu.edu.tw

Molnupiravir is one of the rare orally bioavailable pro-drugs of the nucleoside analog β-D-N4-hydroxycytidine (synonyms: N4-hydroxycytidine; NHC; EIDD-1931) that was used to treat SARS-COV-2; NHC is the active metabolite of molnupiravir and is widely used in broad-spectrum antiviral drugs. NHC is phosphorylated by host kinases to the active intracellular metabolite EIDD-1931-5'-triphosphate (EIDD-2061)[1–3]. The pharmacological mechanism of molnupiravir is similar to that of remdesivir, which targets the RNA-dependent RNA-polymerase enzyme (RdRp) used by the coronavirus for the transcription and replication of its viral RNA genome[4]. NHC is inserted into viral RNA to replace uracil, and while RdRp uses NHC-containing RNA as a template, the enzyme can subsequently incorporate an incorrect nucleotide into the growing RNA strand, leading to mutagenesis[5]. NHC demonstrated poor bioavailability in an animal model; after oral treatment with molnupiravir, NHC was rapidly metabolized in vivo[6,7]. The report published by Merck Pharmaceutics indicates that the bioavailability of molnupiravir is greater than 90%[8], and it has a favorable pharmacokinetic profile and can quickly inhibit the replication of severe acute respiratory syndrome coronavirus 2 (SARS-CoV-2)[9]. A recent study demonstrated that SARS-CoV-2 can cross the BBB and cause damage to the central nervous system and brain[10,11]. However, to date, the biodistribution of molnupiravir and NHC into the brain and the potential mechanism of brain penetration have not been clearly investigated.

Previous reports showed that the effective concentration of NHC against SARS-CoV-2 in an in vitro study was between 0.5 μM and 1 μM (equivalent to 0.13–0.26 μg/mL) in different cell lines[12], and a report showed that the 90% effective concentration (EC$_{90}$) of NHC was 6 μM in the Vero 76 cell line[13]. In and in vivo studies, animals infected with SARS-CoV-2 were treated with molnupiravir by oral administration at doses of 128 and 200 mg/kg in ferrets and hamsters, respectively[14,15]. Additionally, this dose inhibits SARS-COV-2[10,11,16,17]. However, whether the concentration of molnupiravir and NHC is higher than the effective concentration is still unclear.

Drug transporters play a key role in drug absorption, distribution, and excretion by facilitating influx and efflux. They are expressed in many tissues, such as the intestine, liver, kidney, and brain[18]. The equilibrative nucleoside transporter (ENT) is a bidirectional transporter[19], and it mainly transports nucleosides between the blood and endothelium of the brain[20]. A previous report on computational 3D pharmacophore models and in vitro data suggested that increased NHC uptake may occur due to ENT being blocked by nitrobenzylthioinosine (NBMPR), resulting in improved efficacy against SARS-CoV-2[21]. However, there is still no in vivo evidence to demonstrate the effect of modulation of transporters for molnupiravir and NHC in the BBB.

Microdialysis is a minimally invasive sampling technique and can cause transient dysfunction of the biological barrier. It is widely used to collect unbound protein samples from extracellular tissues or organs in vivo[22,23]. The microdialysis probe comprises a semipermeable membrane, and the principle of microdialysis is based on the analyte concentration gradient between the perfusate and the analytes in the extracellular compartment, allowing hydrophilic small molecules to penetrate the membrane. The advantage of microdialysis is that only unbound proteins can pass the exchange region and collect within the dialysate, and the technique does not involve a complicated sample clean-up procedure.

In this study, a multisite microdialysis and a validated ultra-high-performance liquid chromatography-tandem mass spectrometry (UHPLC-MS/MS) are developed to monitor protein-unbound molnupiravir and NHC in the blood and brain. Molnupiravir is rapidly metabolized to NHC and both analytes are detected simultaneously in the dialysates of the blood and brain after administration of molnupiravir. The concentration versus time profile of molnupiravir and NHC in the blood and brain is elucidated. The maximum concentration of NHC in the blood and brain is above the average half-maximum inhibitory concentration of the drug for the treatment of SARS-CoV-2 infection, suggesting a therapeutic effect. After treatment with NBMPR, an inhibitor of ENT, the efflux system of molnupiravir and NHC is blocked and the levels of analytes in the brain are enhanced. The results suggest that the ENT transporter is involved in modulating the efflux system of analytes in the brain.

## Methods

**Chemicals and reagents**. Molnupiravir and NHC were purchased from ChemScene (New Jersey, USA). Nitrobenzylthioinosine (NBMPR) was obtained from Cayman (MI, USA). MS-grade acetonitrile was purchased from J.T. Baker, Inc. (Phillipsburg, NJ). Ammonium acetate was obtained from Sigma–Aldrich Chemicals (St. Louis, MO). Ultrapure water (Millipore, Bedford, MA) was used for sample preparation. A standard stock solution of molnupiravir and NHC (1 mg/mL) was dissolved in acetonitrile and stored at −20 °C for experimental use. The chemical analysis and validation of the method for the analytes are demonstrated in the Supplementary Information.

**Animal experiments and dose administration**. All animal surgery experimental procedures were reviewed and approved by the Institutional Animal Care and Use Committee (IACUC no. 1110210) of the National Yang Ming Chiao Tung University, Taipei, Taiwan. Six-week-old male Sprague–Dawley rats weighing 200 ± 50 g were obtained from the Laboratory Animal Center of National Yang Ming Chiao Tung University. The guideline of animal experiments was followed Guide for the Care and Use of laboratory animals[24] and ARRIVE[25]. Rats had free access to the laboratory rodent diet (PMI Feeds, Richmond, IN, USA) and were given *ad libitum* access to water. The animal facilities were controlled with a 12 h light/dark cycle throughout all feeding processes. The food was removed for 12 h before surgery. Urethane (1 g/kg, i.p.) was administered for anesthesia, and the rats were kept anesthetized during the experimental period; the toe-pinch reflex was used to determine the level of anesthesia. Because the analgesic may influence the pharmacokinetics results, here we only used an anesthetic to reduce pain during surgery. An electric blanket was used to maintain the body temperature during the experiment. A polyethylene tube-50 (PE-50) was placed in the femoral vein for further drug administration. The experimental dosage was determined based on the clinical daily dose (1600 mg/day/human). The bioavailability of molnupiravir is 90%[26], and the transfer constant to a rat dosage of molnupiravir was determined (147 mg/kg, i.v.). In the experiment, considering that molnupiravir is taken once every 12 h, the dose we used was 100 mg/kg.

**Microdialysis experiment**. The microdialysis equipment comprised a microinjection pump (CMA/400, CMA, Stockholm, Sweden), a microfraction collector (CMA/142, CMA), and the dialysate collection probe, which were designed and applied in our laboratory[27,28]. The microdialysis probe comprised a concentric silica capillary with a semipermeable dialysis membrane (Spectrum, New Brunswick, NJ, USA) in which the fiber had an inner diameter of 200 μm with a molecular weight cutoff of 13 kDa, and substance exchange occurred due to the concentration gradient. Due to the position of each sampling target, the lengths of the blood and brain dialysis membrane were 1.2 cm

and 0.7 cm, respectively. Initially, the blood microdialysis probe was inserted on the right side of the jugular vein in the direction of the heart, and the perfusate was maintained with an anticoagulant citrate dextrose (ACD) solution comprising 3.5 mM citric acid, 7.5 mM sodium citrate and 13.6 mM D-(+)-glucose at a flow rate of 2.0 μL/min. The left femoral vein was catheterized with PE-50 for intravenous drug injection. For brain microdialysis sampling, rats were fixed on a stereotaxic instrument (David Kopf Instruments, Tujunga, CA, USA) for the implantation of the microdialysis probe. After mounting with a stereotaxic instrument, the hole was drilled in the skull with a pen-type grinder, and a brain probe was implanted at the striatum site (+0.2 mm anteroposterior, +3.0 mm mediolateral and −7.5 mm dorsoventral to bregma) from the scull according to the guide The Rat Brain in Stereotaxic Coordinates[29]. The perfusate was an ACD solution with a flow rate of 2.0 μL/min. A minimum one hr stabilization period should be required after implantation of the microdialysis probe, molnupiravir (100 mg/kg, i.v.) was administered through the femoral vein, and the other group was concomitantly administered molnupiravir (100 mg/kg, i.v.) and NBMPR (10 mg/kg, i.v.). Dialysates of the blood and brain were collected every 20 min for 6 h using a microfraction collector (CMA/142) and kept at −20 °C for further UHPLC–MSMS/MS analysis. All drug concentrations were corrected by recovery percentage.

**Statistics and reproducibility.** Profiling solution software (version 1.1; Shimadzu, Kyoto, JPN) was used to evaluate the chromatogram data. WinNonlin Standard Edition software (version 1.1; Scientific Consulting Inc., Apex, NC, USA) was used to calculate the main pharmacokinetic parameters with a one-compartment model of molnupiravir in blood selected based on Akaike's Information Criterion (AIC)[30] and the noncompartmental model of molnupiravir in the brain and NHC in the blood and brain. The other parameters included the area under the concentration curve (AUC), initial drug concentration ($C_0$), maximum concentration of the drug ($C_{max}$), half-life ($t_{1/2}$), clearance (CL), mean residence time (MRT), biotransformation ratio of molnupiravir to NHC calculated using [$AUC_{NHC}/AUC_{molnupiravir}$] × 100%, and biodistribution ratio of blood to the brain calculated using [$AUC_{brain}/AUC_{blood}$] × 100%. The drug concentration-time curves were drawn by SigmaPlot (version 10.0; Systat Statistics, London, UK). Statistical analyses were performed using SPSS Statistics (version 22.0, IBM Corp., Armonk, NY, USA). Statistical differences were determined using one-way analysis of variance (ANOVA) and Tukey's post hoc test, and the $AUC_{brain}/AUC_{blood}$ of molnupiravir and NHC in the group given molnupiravir (100 mg/kg, i.v.) alone were determined using Student's $t$-test with post hoc Tukey's HSD test. The alpha criterion for significance was set at 0.05.

**Reporting summary.** Further information on research design is available in the Nature Portfolio Reporting Summary linked to this article.

## Results

**Optimization of UHPLC–MS/MS conditions.** The UHPLC–MS/MS system was developed to analyze molnupiravir and NHC levels in dialysate using the positive MRM-ESI mode. The analytical method was in accordance with the validation guidelines released by the US Food and Drug Administration in 2018[31] and provided high sensitivity and selectivity for the quantification of molnupiravir and NHC levels. After optimization, mass transitions of molnupiravir and NHC were observed at 330.2 to 128 (m/z) and 260 to 128.05 (m/z), respectively. The collision energies were −15 eV and −11 eV for molnupiravir and NHC,

respectively (Supplementary Fig. S1). The lower limits of quantification for molnupiravir and NHC in rat dialysates were 2.5 and 10 ng/mL, respectively. Analytical method validation and microdialysis recovery results are presented in Supplementary Tables S1–S5 and Supplementary Data 1.

To improve chromatographic separation, ammonium acetate (2 mM) was added to the aqueous phase, and the pH value was adjusted to 4.3 by acetic acid. The organic phase affected the detection intensity and retention time of molnupiravir and NHC; here, we tested the volume ratio of methanol and acetonitrile to optimize our experiment, and we observed substantially better signal intensity and separation when acetonitrile was used than when methanol was used as the organic phase. The retention times for molnupiravir and NHC were 1.25 and 5.70 min, respectively, and the representative MRM chromatograms did not show an interference signal in the blank matrix during the retention of the molnupiravir and NHC (Fig. 1).

**Biotransformation and brain distribution of molnupiravir and NHC.** Based on the validated analytical system described above, molnupiravir and NHC were detected in the blood (Fig. 2a) and brain (Fig. 2b) in rats after molnupiravir administration (100 mg/kg, i.v.); rapid biotransformation and brain distribution of molnupiravir and NHC were observed during the first 20 min, and rapid biodistribution into the brain occurred with $C_{max}$ at 40 min. Molnupiravir was detected only during the first 100 min after administration, indicating a short elimination time and rapid metabolism, and $t_{1/2}$ ranged from 12 ± 2 (min) in the blood. Furthermore, the elimination rate of NHC was slow, and NHC could be detected in dialysates of the blood and brain at 360 min (Fig. 2). The elimination half-life of NHC from the blood and brain was 95 ± 9 and 105 ± 12 min, respectively (Table 1 and Supplementary Data 2).

The area under the concentration curve (AUC) of molnupiravir and NHC in the blood was 4214 ± 688.7 and 7228 ± 339.9 min μg/mL, respectively, and was 13.39 ± 0.96 and 63.38 ± 3.12 min μg/mL, respectively, in the brain. The biotransformation and biodistribution of molnupiravir to NHC were evaluated using the $AUC_{NHC}/AUC_{molnupiravir}$ ratio from the blood and brain, which was 1.71 ± 0.08 and 4.50 ± 0.22, respectively. The results demonstrated that the biodistribution ratio in the brain was significantly higher than that in the blood. The biodistribution ratios ($AUC_{brain}/AUC_{blood}$) of molnupiravir and NHC in the brain were 0.3 ± 0.02% and 0.8 ± 0.04%, respectively (Table 1). These results demonstrated that NHC crossed the BBB into the brain, and the transfer ratio was between 0.3–0.8%.

**Mechanism of BBB transportation of molnupiravir and NHC.** To investigate the mechanism of BBB transport by which molnupiravir and NHC cross into the brain, an equilibrative nucleoside transporter inhibitor, NBMPR (10 mg/kg, i.v.) was concomitantly administered with molnupiravir (100 mg/kg, i.v.). The results showed that the AUCs of molnupiravir and NHC in blood were 4191 ± 623.5 and 12410 ± 698.7 min μg/mL, respectively, and the AUCs in the brain were 32.25 ± 4.09 and 190.5 ± 32.74 min μg/mL, respectively. Compared with molnupiravir alone group, the AUCs and $C_{max}$ of molnupiravir and NHC in brain are significantly increased ($p < 0.05$) (Table 1).

The biotransformation and biodistribution ($AUC_{NHC}/AUC_{molnupiravir}$) of molnupiravir in the blood and brain were 2.96 ± 0.23 and 5.83 ± 1.00, respectively. Compared to molnupiravir alone, NBMPR did not significantly affect molnupiravir biotransformation and biodistribution in the blood and brain (Table 1). The biodistribution ratios ($AUC_{brain}/AUC_{blood}$) of molnupiravir and NHC were 0.007 ± 0.0001 and 0.015 ± 0.002,

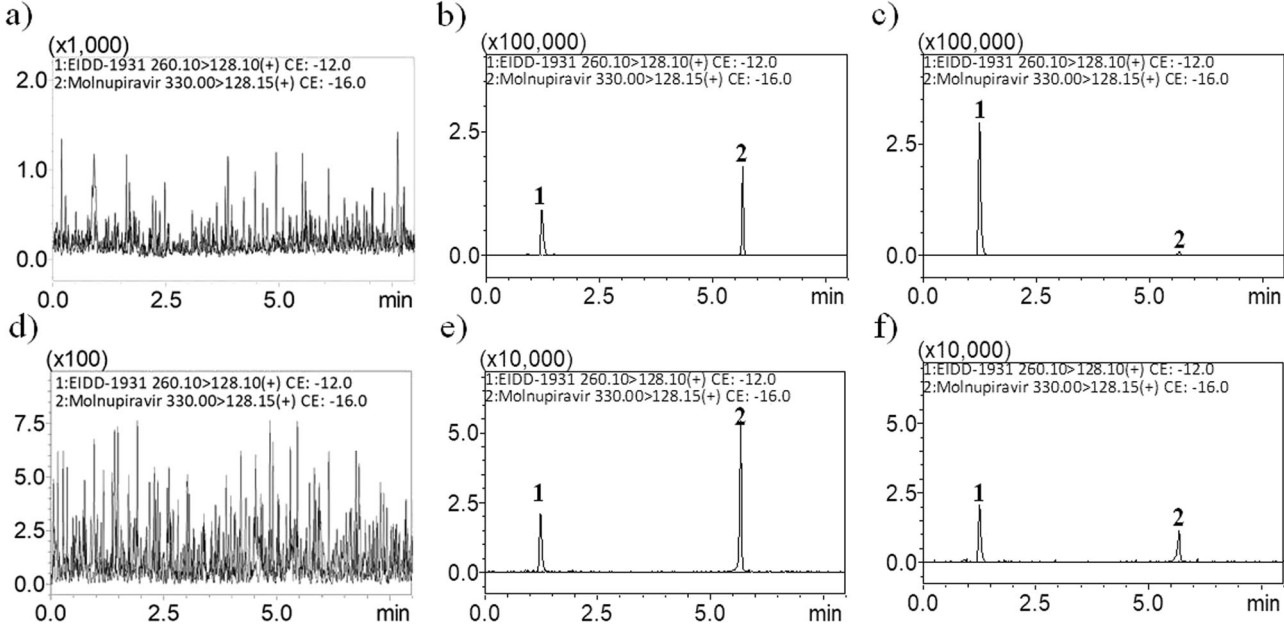

**Fig. 1 Representative MRM chromatograms of blood and brain dialysates. a** blank blood dialysate, **b** blank blood dialysate spiked with β-D-N4-hydroxycytidine (NHC; 250 ng/mL) and molnupiravir (100 ng/mL), **c** blood dialysate sample containing NHC (758.5 ng/mL) and molnupiravir (4.6 ng/mL) collected at 140 min after molnupiravir administration (100 mg/kg, i.v.), **d** blank brain dialysate, **e** blank brain dialysate spiked with NHC (50 ng/mL) and molnupiravir (25 ng/mL), **f** brain dialysate sample containing NHC (47.2 ng/mL) and molnupiravir (5.52 ng/mL) collected 60 min after molnupiravir administration (100 mg/kg, i.v.). Peak 1: NHC, Peak 2: molnupiravir.

respectively. Compared with molnupiravir alone group, the biodistribution ratio of molnupiravir and NHC of the brain is significantly increased ($p < 0.05$) (Table 1). It suggests that NBMPR significantly improved the penetration of molnupiravir and NHC into the brain (Table 1).

## Discussion

The multiple microdialysis technique used in this study was in compliance with the 3Rs principle (replacement, reduction, and refinement) for experimental animal studies, which reduced the number of animals needed to collect samples, measure the protein-unbound drug in the blood vessel and extracellular space of the brain, and continuously monitor the kinetic changes of the analyte. To monitor neurotransmitters and their metabolites with microdialysis, a minimum 1-h stabilization period should be required to balance endogenous monoamine levels back to a stable baseline[32]. Due to a serious mass matrix effect that significantly affected the NHC signal when the Ringer solution was used, an anticoagulant citrate dextrose (ACD) solution was used as a perfusate during blood and brain dialysis to decrease the matrix effect. The number of animals was lower using the microdialysis system than when conventional methods for collecting biological samples were used. Furthermore, the sample collected by microdialysis does not require the sample preparation and can be injected directly into UHPLC–MS/MS, which is more convenient for analysis[33,34].

The results demonstrated that the concentration of molnupiravir in rat blood decreased rapidly after molnupiravir administration, which is consistent with the previous report of a short half-life ($t_{1/2}$) of molnupiravir and rapid metabolism to NHC through hydrolysis by carboxylesterases in the intestine and liver[26]. To prevent disturbance of the drug via the first-pass effect associated with oral administration and focus the study on the BBB, intravenous administration was used in this experiment. The dose of molnupiravir (100 mg/kg, i.v.) used in this experiment was based on the clinical dose of molnupiravir (1600 mg/day, p.o.) multiplied by the bioavailability[2,8] to obtain the experimental dose.

After administration of molnupiravir (100 mg/kg, i.v.), the $C_{max}$ of molnupiravir and NHC in the blood was $219.0 \pm 45.58$ and $148.4 \pm 7.56$ μg/mL, respectively, and the $C_{max}$ in the brain was $0.35 \pm 0.03$ and $0.70 \pm 0.05$ μg/mL, respectively. In 2020, Sheahan et al. reported that NHC had potent antiviral effects with an average half maximal inhibitory concentration ($IC_{50}$) of 0.08 μM to 0.3 μM (equivalent to 20.8–78 ng/mL)[12]. Our results suggested that NHC penetrated the BBB, and the concentration was within or above therapeutic concentrations approximately 300 min after molnupiravir administration (100 mg/kg, i.v.). These results were also consistent with a clinical report in which molnupiravir was administered 800 mg twice daily for patients with SARS-CoV-2 infection, the concentrations of NHC in the plasma, saliva, nasal samples, and tear samples were maintained within or above the 90% $EC_{90}$ of 0.5–1 μM, which is approximately equivalent to 0.13–0.26 μg/mL of NHC, and the treatment showed protection against SARS-CoV-2 infection[6,7]. Another report with single and multiple ascending doses demonstrated that NHC appeared rapidly in plasma, with a median $T_{max}$ of 1 h and $C_{max}$ that were all effectively above the $EC_{90}$ after dosing[2,7].

Due to the low plasma concentration reflecting the poor human bioavailability of NHC when NHC was administered alone[6] and the high extent of biotransformation of molnupiravir to NHC, the intravenous route was ideal for the administration of molnupiravir to investigate the possible penetration mechanism of NHC through the BBB in this study. Furthermore, according to a report by the European Medicines Agency (EMA), molnupiravir and NHC-related material are metabolized to endogenous pyrimidine nucleosides (uridine and/or cytidine), and NHC-triphosphate is retained in the body with a long elimination half-life[26]. In previous reports, the $T_{max}$ of NHC was found to be approximately 1–2 h[2,7,26], which is consistent with our results of $95 \pm 9$ and $105 \pm 12$ min in the blood and brain, respectively (Table 1), after administration of molnupiravir (100 mg/kg, i.v.).

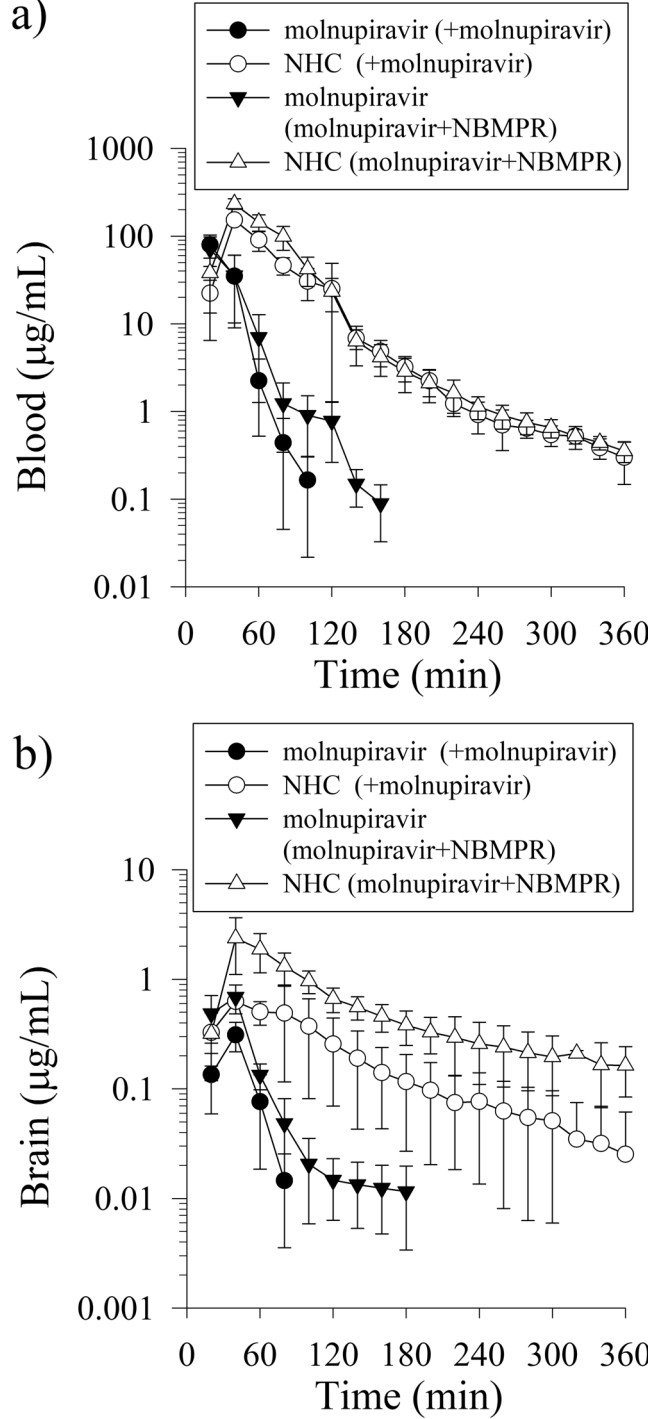

**Fig. 2 Protein-unbound molnupiravir and NHC in the rat blood and brain. a** Concentration-time curves of molnupiravir and β-D-N4-hydroxycytidine (NHC) in blood and **b** concentration-time curves of molnupiravir and NHC in brain after molnupiravir administration (100 mg/kg, i.v.) alone and administration of molnupiravir (100 mg/kg, i.v.) + NBMPR (10 mg/kg, i.v.). Data are expressed as the mean ± SD ($n = 6$); nitrobenzylthioinosine (NBMPR; an inhibitor of ENT). The symbols ● and ○ represent the concentrations of molnupiravir and HNC, respectively, in administered molnupiravir (100 mg/kg, i.v.) alone. The symbols ▼ and △ represent the concentrations of molnupiravir and HNC, respectively, in concomitantly administered molnupiravir (100 mg/kg, i.v.) and NBMPR (15 mg/kg, i.v.).

The concentration versus time curves of molnupiravir and NHC in the brain showed declining trends similar to those in the blood (Fig. 2). The molnupiravir concentration reached $T_{max}$ at 40 min with a maximum concentration of $0.35 \pm 0.03$ μg/mL and was rapidly metabolized until undetectable after 80 min. This result indicated a delay in the entry of molnupiravir into the brain and a rapid decline of its concentration during the dose regimen. However, microdialysis has some limitations when the microdialysis probe is inserted into the brain for sampling. The surgery process and the implantation of the microdialysis probe can affect the integrity of the blood–brain barrier and the recovery time of the surgery process and the stabilization period after the implantation of the microdialysis probe. Due to the invasiveness of implanted probes in the brain, an abnormal release of neurotransmitters could be disturbed. Generally, a recovery period for surgery longer than 24 h is considered satisfactory for monitoring neurotransmitter release in a conscious and freely moving rat. Subsequently, on the day of the experiment, the dummy cannula was removed from the guide cannula and replaced by the real microdialysis probe and following a 60-min[35] and 2-h[36] period for stabilization. For the pharmacokinetic study, rats were fixed on a stereotaxic instrument to implant the microdialysis probe, prior to drug administration a stabilization period of 30 min[37], 1 h[35,38], 1.5 h[39] and 2 h[40] was generally applied. In our experiment, consider the experiment without being treated with transporter inhibitor, the $AUC_{brain}/AUC_{blood}$ ratios of molnupiravir and NHC were 0.3% and 0.8%, respectively, but after treatment with NBMPR (10 mg/kg, i.v.) the $AUC_{brain}/AUC_{blood}$ ratios were increased to 0.7% and 1.5%, respectively (Table 1), suggesting that the ENT transporter has been inhibited and maintains the integrity function of the BBB.

Additionally, the NHC concentration showed a $T_{max}$ range of $43 \pm 3$ min with a maximum concentration of $0.70 \pm 0.05$ μg/mL, which means that NHC also exhibited delayed crossing of the BBB. Furthermore, our results showed a low brain distribution ratio ($AUC_{brain}/AUC_{blood}$) of 0.3 and 0.8% for molnupiravir and NHC (Table 1), respectively, which is consistent with the reports provided by the EMA[2,26]. Despite the low BBB distribution, the $C_{max}$ of NHC ($0.70 \pm 0.05$ μg/mL) is still much higher than the $IC_{50}$ of NHC, which ranges from 0.08 μM to 0.3 μM (equivalent to 20.8–78 ng/mL) at up to approximately 360 min[12].

In 2023, the study published by Saleh et al. used LeiCNS-PK3.0, a physiologically based pharmacokinetic (PBPK) model, to predict whether the concentration of molnupiravir in the brain can reach the effective concentration[41]. They reported that the dosage needs to be 4000 mg twice daily and can be effective in the brain. Previous data indicated that the EC50 and EC90 values for treating infections with the delta SARS-COV-2 strain are 1.43 and 4.65 μM (equivalent to 471.9 and 1534 ng/mL), respectively, and the EC50 value for treating infections with the omicron SARS-COV-2 strain is 0.25 μM (equivalent to 82.5 ng/mL). In our study, the drug concentration in blood can reach both the EC50 and EC90 for treating infections with delta and the $EC_{50}$ for treating infections with omicron; however, the concentration of NHC in the brain could only reach the $EC_{90}$ for treating infections with delta for a short time if administered with molnupiravir at 100 mg/kg.

Based on an EMA report, neither molnupiravir nor NHC were found to be substrates of human MDR1 P-glycoprotein (P-gp) or BCRP, and the substrate of the human nucleoside transporters CNT1, CNT2, CNT3 was found to be the transporter of NHC in vitro[26]. Another study found that the transporter of NHC was equilibrative nucleoside transporters 1 and 2 (ENT1 and ENT2) in the blood-testis barrier through in vitro experiments and

**Table 1 Pharmacokinetic parameters of molnupiravir and NHC in the rat blood and brain after treatment with molnupiravir (100 mg/kg, i.v.) or molnupiravir (100 mg/kg, i.v.) + NBMPR (10 mg/kg, i.v.).**

| Parameter | Molnupiravir (100 mg/kg, i.v.) | | | | Molnupiravir (100 mg/kg, i.v.) + NBMPR (10 mg/kg, i.v.) | | | |
|---|---|---|---|---|---|---|---|---|
| | Blood | | Brain | | Blood | | Brain | |
| | Molnupiravir | NHC | Molnupiravir | NHC | Molnupiravir | NHC | Molnupiravir | NHC |
| $AUC_{last}$ (min µg/mL) | 4214 ± 688.7 | 7228 ± 339.9 | 13.39 ± 0.96 | 61.38 ± 3.12 | 4191 ± 623.5 | 12410 ± 698.7 | 32.25 ± 4.09 | 190.5 ± 32.74 |
| $C_{max}$ (µg/mL) | 219.0 ± 45.58 | 148.4 ± 7.56 | 0.35 ± 0.03 | 0.70 ± 0.05 | 216.1 ± 36.03 | 233.6 ± 13.07 | 0.68 ± 0.09 | 2.14 ± 0.46 |
| $t_{1/2}$ (min) | 14 ± 1 | 95 ± 9 | 9 ± 1 | 105 ± 12 | 14 ± 1 | 72 ± 8 | 49 ± 19 | 182 ± 77 |
| $T_{max}$ (min) | - | 40 | 36 ± 3 | 43 ± 3 | - | 40 | 40 | 43 ± 3 |
| CL (mL/min/kg) | 28.32 ± 4.91 | - | - | - | 27.30 ± 3.93 | - | - | - |
| MRT (min) | 20 ± 2 | 63 ± 1 | 33 ± 3 | 86 ± 2 | 20 ± 3 | 60 ± 1 | 33 ± 2 | 115 ± 10 |
| $AUC_{NHC}/AUC_{molnupiravir}$ | - | 1.71 ± 0.08 | - | 4.50 ± 0.22 | - | 2.96 ± 0.23 | - | 5.83 ± 1.00 |
| $AUC_{brain}/AUC_{blood}$ | - | - | 0.003 ± 0.0002 | 0.008 ± 0.0004 | - | - | 0.007 ± 0.0001 | 0.015 ± 0.002 |

*AUC* area under curve the concentration-time curve, $t_{1/2}$ half-life, $C_{max}$ concentration maximum, *CL* clearance, *MRT* mean residence time.
$AUC_{brain}/AUC_{blood}$ represents the rat blood-to-brain transfer ratio. Data have been expressed as the mean ± SEM ($n = 6$); nitrobenzylthioinosine (NBMPR; an inhibitor of ENT).

computer simulations[21]. To date, the mechanisms underlying the transport of molnupiravir and NHC in the brain are not fully understood. In this study, we used NBMPR, an ENT inhibitor, to block the transporters of molnupiravir and NHC in experimental rats. The results showed that the blood-to-brain ratios of molnupiravir and NHC concentration increased significantly from 0.3% to 0.8% and 0.7% to 1.5%, respectively, in the brain (Table 1). These results suggested that the bidirectional influx and efflux function of the ENT transporter was blocked to cause accumulation of molnupiravir and NHC in the brain, confirming a previous report that molnupiravir and NHC were modulated by nucleoside transporters (CNT and ENT)[19,42,43]. When the ENT channel was blocked, the efflux system was also blocked, and molnupiravir and NHC accumulated in the brain through CNTs.

In conclusion, an optimized UHPLC−MS/MS method coupled with microdialysis for monitoring molnupiravir and NHC in rat blood and brain dialysates was developed. Molnupiravir was rapidly metabolized to NHC, and both analytes were simultaneously detected in the dialysates of the blood and brain. After treatment with the ENT inhibitor, the efflux system of molnupiravir and NHC was blocked, and the levels of analytes were enhanced. Through multiple microdialysis of the blood and brain, the dynamic profile of the biotransformation of molnupiravir into the NHC and their distribution into the brain were elucidated. The ENT transporter was found to be involved in the regulation of the efflux system of both molnupiravir and NHC in the brain.

### Data availability

All relevant data supporting the findings of this study are included in this published article and its Supplementary Information. The source data underlying the analytical method validation and pharmacokinetics are included in Supplementary Data 1 and Supplementary Data 2, respectively. These data are available from the corresponding author upon reasonable request.

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

## Acknowledgements

This study was conducted as part of the PhD dissertation of Chun-Hao Chang. This study was supported by research grants from the National Science and Technology Council of Taiwan (NSTC 111-2113-M-A49-018 and NSTC 112-2321-B-A49-005) and a graduate student scholarship from the College of Medicine, National Yang Ming Chiao Tung University, Taipei, Taiwan.

## Author contributions

Methodology: C.H.C., W.H.L., L.Y.; Investigation: C.H.C., W.H.L., L.Y., T.Y.L., W.Y.P.; Visualization: T.Y.L., W.Y.P.; Funding acquisition: M.H.Y., T.H.T.; Project administration: M.H.Y., T.H.T.; Supervision: T.H.T.; Writing—original draft: C.H.C., T.H.T.; Writing—review & editing: C.H.C., T.H.T.; All authors have read and approved the final version of the manuscript.

## Competing interests

The authors declare no competing interests.
