## [Peer review file · Communications Medicine]

Reviewers' comments:

Reviewer #1 (Remarks to the Author):

The manuscript by Chang et al. investigates the blood-brain barrier (BBB) penetration of a potential COVID-19 antiviral molnupiravir in rats. They utilize dual-probe in vivo microdialysis approach for extracting the protein unbound fractions of the prodrug molnupiravir and its active form β -D-N4-hydroxycytidine (NHC) in blood and in brain extracellular fluid (i.e., on both sides of the BBB). Drug concentrations in dialysates were measured by a liquid chromatography-tandem mass spectrometry method. Furthermore, the authors hypothesize that molnupiravir and NHC are actively transported at the BBB by the equilibrative nucleoside transporter (ENT). The role of ENT in brain transport of molnupiravir and NHC is validated by blocking it using an inhibitor nitrobenzylthioinosine (NBMPR). The authors found increased brain penetration of molnupiravir and NHC following the ENT inhibition, leading to the conclusion that ENT effluxes both compounds out of the brain.

The topic of the research is important. The knowledge about brain viral load after SARS-CoV-2 infection is increasing and there certainly is a need for brain-penetrating antivirals. Molnupiravir has shown efficacy against pulmonary SARS-CoV-2 infection and is one of the rare perorally administered drug therapies for COVID-19. Thus, this manuscript addresses an important question and the microdialysis data is novel. However, the manuscript suffers from rather optimistic data interpretation, severe methodological problems, and incomplete scientific presentation. The authors should consider the following points to improve the quality of their work:

- 1) It is claimed in the manuscript that ENTs would efflux molnupiravir and NHC at the BBB (row 272, 285) and this can be modulated by an ENT inhibitor. However, according to the literature, and also stated by the authors elsewhere (row 70), ENTs are responsible for the uptake of nucleosides (and nucleoside-derived drugs) from blood to the brain and further distribution to the cells. Thus, inhibition of this uptake route would, to my mind, lead to decreased brain drug delivery, not increased as stated. In fact, a recent study reported that NHC accumulation in ENT-expressing HeLa cells decreased in the presence of NBMPR, suggesting that other carrier-mediated pathways are involved in the uptake of NHC and molnupiravir (doi: 10.1124/molpharm.121.000333). In the graphical abstract, there appears to be an attempt to explain the altered pharmacokinetics by increased uptake via CNT transporters, but this has not been discussed in the manuscript. The authors should re-evaluate their data and discussion regarding the NBMPR part of the study.
- 2) The microdialysis method raises several concerns. It may be that because of below mentioned methodological problems the brain penetration of molnupiravir is an artefact. Molnupiravir is unstable in plasma and has a very low potential for BBB penetration.
 - a. It is well known that the insertion of the microdialysis probe into the brain causes a transient dysfunction of the BBB. It takes, at least, several hours to restore the normal function. One hour balancing period used in this study is simply too short and will probably lead to overestimation of drug brain penetration. This point should be discussed.
 - b. The perfusate composition (ACD solution) is unusual, especially for brain microdialysis. Normally, the perfusate composition tries to mimic the extracellular fluid composition and pH as closely as possible. ACD solutions are acidic and missing key ions of the extracellular fluid, which will affect the pH and ion composition of the tissue surrounding the probe. This may compromise the integrity of the BBB. This is a clear limitation of the study and should be discussed.
 - c. The in vitro probe recovery calculation is problematic and may lead to erroneous data manipulation. This is especially true when two very different tissues (blood and brain) are being compared (AUC ratios), and because the perfusate composition differs so much from that of the investigated tissues. The in vivo recoveries should be determined and used for recovery calculations. The authors should also highlight in the text and tables that the drug concentrations are corrected with recovery values.
 - d. A minor comment: are the stereotaxic coordinates calculated from the top of the skull or dura mater? Is it the tip of the probe that is extending to -7,5 mm (from dura or skull)? Please specify these details.

- 3) The authors make rather optimistic translational interpretations: they justify the dose of molnupiravir (100 mg/kg i.v.) by human clinical dose (1600 mg/day p.o.) "multiplied by the bioavailability". However, the species differences are not considered in this calculation. Is the dose in accordance e.g. to the FDA guideline for calculating the human equivalent dose based on the body surface area of the species? Furthermore, it is important to keep in mind that the microdialysis method measures the extracellular unbound drug concentrations in a fraction of time (20 min in this case), and the measured concentrations do not represent exact concentrations at specific time points as often misinterpreted. Although the authors justify the use of i.v. route, the translational value of the findings would increase if molnupiravir had been administered perorally as in patients.
- 4) Scientific presentation is incomplete, and some important literature is missing. Some examples:
- Methods section is presented twice (chapters 3 and 5), why is that?
 - graphical abstract is missing the blood microdialysis
 - a more thorough description of molnupiravir pharmacology should be given in the Introduction (molnupiravir – NHC – triphosphated nucleoside analog EIDD-2061)
 - There are recent reports on the pharmacokinetics of EIDD-1931 and EIDD-2061 that should be incorporated into the manuscript (e.g., <https://doi.org/10.1016/j.antiviral.2019.104597>; doi: 10.1016/j.ejps.2022.106345). Furthermore, it should be mentioned that molnupiravir is one of the rare orally administered drugs against COVID-19.
 - row 54: what does "can quick clearance" mean?
 - row 60: effective concentration of NHC against what?
 - row 62: "molnupiravir was treated", how a treatment can be treated?
 - row 65: effective concentration against what and where?
 - row 73: what does modulation mean here?
 - row 115: based on the time-concentration curves (fig 2) the compounds seem to follow a two-compartment model with distinct distribution and elimination phases, thus, please specify which elimination half-life (alpha, beta) has been calculated
 - rows 120-123: AUC ratio of molnupiravir:NHC is miscalculated ($7228/4214 = 1.71$, not 2.10 as stated)?
 - row 124: it cannot be concluded that the biotransformation ratio of molnupiravir to NHC was significantly higher in the brain than in the blood; NHC in the brain is most likely released in the periphery and transported into the brain, not "biotransformed" there.
 - rows 126-127: check the numbers (AUC ratio 0.003 and 0.008, i.e., transfer ratios 0.2 and 0,9%?)
 - row 162: how was the depth of anesthesia controlled?
 - row 227: how does the IC₅₀ of NHC suggest that NHC penetrated the BBB? Rephrase.
 - rows 230-232: rephrase to clarify the point.
 - row 271: what aspect of molnupiravir and NHC was modulated by the transporter? Please clarify and rephrase.
 - The authors could better highlight that the IC₅₀ or EC₉₀ values determined in in vitro cellular assays are usually protein unbound concentrations measured from the cell medium, and these concentrations are comparable with those measured by in vivo microdialysis (extracellular unbound concentrations).
 - discussion on rows 275-280 is purely speculative, not based on the data, and the point is unclear. Molnupiravir has been shown to be unstable in plasma, thus, it is plausible that most of the NHC found in the brain is released in the periphery. I recommend removing this paragraph.
 - The IC₅₀ values in figure 2 should be removed (because they are not determined in this study). Also, the figure legends are difficult to interpret (what does "molnupiravir (+molnupiravir)" mean?). Figure text states that values represent mean ± SEM, but in chapter 3.4. it is said that data have been expressed as mean ± SD?
 - Some of the statistical methods (Mann-Whitney U-test) that are mentioned in the footnote of table 1 are not mentioned in the methods section. Mann-Whitney U-test cannot be combined with Tukey's post hoc test. Why parametric and nonparametric tests have been mixed, have the authors tested the normal distribution of the data? The use of different tests should be justified.
 - English language needs revision throughout the manuscript

Reviewer #2 (Remarks to the Author):

The authors present a new methodology to detect and quantify blood and brain dialysate levels of molnupiravir (prodrug) and NHC (active metabolite). The data is presented in order to combat SARS-CoV-2, as a means to target viral transcription and replication within the brain since it has been shown SARS-CoV-2 can enter the brain. The authors quantify the amount of molnupiravir and NHC in the brain interstitial fluid and blood as a means to identify transport pharmacokinetics. However, as NHC is a metabolite of molnupiravir, they cannot be certain the NHC detected in brain occurred from BBB transport or metabolism of molnupiravir once present in brain. They also try to identify the transporter responsible, finding inhibition of ENT results in increased drug levels within the brain, indicative of a role of this transporter in efflux. However, the authors state this transporter is primarily involved in blood to endothelial cell transport and do not provide adequate discussion regarding the discrepancy with their findings. The transporter responsible for blood to brain still remains unknown.

Major Comments:

1. Are the authors really investigating the transporter modulation as stated in the title (i.e., can they prove they are directly modulating the transporters for molnupiravir and NHC)?
2. Please see the very recent paper published in European Journal of Pharmaceutical Sciences by Saleh et al on the prediction of BBB transport and efficacy of Molnupiravir and discuss in the Discussion: The PBPK LeICNS-PK3.0 framework predicts Nirmatrelvir (but not Remdesivir or Molnupiravir) to achieve effective concentrations against SARS-CoV-2 in human brain cells
3. Line 136-141: I do not think the authors can identify the biotransformation of molnupiravir to NHC occurring in the brain using their methods. How can they correct for the amount of NHC transported across the BBB?
4. Line 250-251: Saturability of molnupiravir was not investigated.
5. Lines 253-254: How is it known NHC is actually crossing the BBB and not being converted from molnupiravir once present in the brain to NHC?
6. Figure 1: The complete description of the samples in C are not described (i.e., how long after molnupiravir administration)
7. As ENT inhibition increases drug levels in brain, indicating a role in brain efflux of the drugs, please discuss or amend the introduction that states ENTs are primarily involved in transport from blood to the endothelium (and not vice versa) and why the results presented here differ.

Minor Comments

1. Grammatical errors: Lines 77-79, 130-132
2. 127, 219, 250, 254: blood-brain barrier has already been defined and abbreviated (BBB)
3. Is line 206-210 critical for the first sentence of the Discussion?
4. Line 275-280: difficult to follow text; please rephrase

Manuscript number: COMMSMED-23-0031-T

Transporter modulation of molnupiravir and β -d-N4-hydroxycytidine at the blood–brain barrier in rats

Chun-Hao Chang¹, Wen-Ya Peng¹, Wan-Hsin Lee¹, Ling Yang¹, Tung-Yi Lin¹, Muh-Hwa Yang², Tung-
Hu Tsai^{1,3,4*}

Reviewer comment:

Reviewer #1:

1) It is claimed in the manuscript that ENTs would efflux molnupiravir and NHC at the BBB (row 272,
285) and this can be modulated by an ENT inhibitor. However, according to the literature, and also
stated by the authors elsewhere (row 70), ENTs are responsible for the uptake of nucleosides (and
nucleoside-derived drugs) from blood to the brain and further distribution to the cells. Thus, inhibition
of this uptake route would, to my mind, lead to decreased brain drug delivery, not increased as stated.
In fact, a recent study reported that NHC accumulation in ENT-expressing HeLa cells decreased in the
presence of NBMPR, suggesting that other carrier-mediated pathways are involved in the uptake of
NHC and molnupiravir (doi: 10.1124/molpharm.121.000333). In the graphical abstract, there appears
to be an attempt to explain the altered pharmacokinetics by increased uptake via CNT transporters, but
this has not been discussed in the manuscript. The authors should re-evaluate their data and discussion
regarding the NBMPR part of the study.

Response: Thank you for the comment. The unclear statement has been revised accordingly.

The equilibrative nucleoside transporter (ENT) is a bidirectional transporter (Podgorska et al., 2005),
and it mainly transports nucleosides between the blood and endothelium of the brain (Anderson et al.,
1999). A previous report on computational 3D pharmacophore models and in vitro data suggested that
increased uptake of NHC may occur due to ENT being blocked by nitrobenzylthioinosine (NBMPR),
resulting in improved efficacy against SARS-CoV-2 (Miller et al., 2021) (doi:
10.1124/molpharm.121.000333). However, there is still no in vivo evidence to demonstrate the
modulation of the transporters for molnupiravir and NHC in the BBB (page 3, line 71-77).

In this study, we used NBMPR, an ENT inhibitor, to block the transporters of molnupiravir and NHC
in experimental rats. The results showed that the blood-to-brain ratios of molnupiravir and NHC
concentration increased significantly from 0.3% to 0.8% and 0.7% to 1.5%, respectively, in the brain
(Table 1). These results suggested that the bidirectional influx and efflux function of the ENT
transporter was blocked to cause accumulation of molnupiravir and NHC in the brain, confirming a
previous report that molnupiravir and NHC were modulated by nucleoside transporters (CNT and ENT)
(Molina-Arcas et al., 2009; Podgorska et al., 2005; Vlachodimou et al., 2019). When the ENT channel

was blocked, the efflux system was also blocked, and molnupiravir and NHC accumulated in the brain
through CNTs (page 7, line 228-234).

Based on an EMA report, neither molnupiravir nor NHC were found to be substrates of human MDR1
P-glycoprotein (P-gp) or BCRP, and the substrate of the human nucleoside transporters CNT1, CNT2,
CNT3 was found to be the transporter of NHC in vitro (European Medicines Agency, 2022). Another
study found that the transporter of NHC was equilibrative nucleoside transporters 1 and 2 (ENT1 and
ENT2) in the blood-testis barrier through in vitro experiments and computer simulations (Miller et al.,
2021) (doi: 10.1124/molpharm.121.000333) (page 7, line 221-225).

2) The microdialysis method raises several concerns. It may be that because of below mentioned
methodological problems the brain penetration of molnupiravir is an artefact. Molnupiravir is unstable
in plasma and has a very low potential for BBB penetration.

a. It is well known that the insertion of the microdialysis probe into the brain causes a transient
dysfunction of the BBB. It takes, at least, several hours to restore the normal function. One hour
balancing period used in this study is simply too short and will probably lead to overestimation of drug
brain penetration. This point should be discussed.

Response: Microdialysis is a minimally invasive sampling technique and may cause transient
dysfunction of the biological barrier. It is widely used to collect unbound protein samples from
extracellular tissues or organs in vivo (de Lange et al., 1997; Tsai, 2003) (page 3, line 79-81).

The question of the maintenance of the integrity of the BBB after insertion of the microdialysis probe
has been addressed in a number of studies. Based on our previous study in which neurotransmitters
and their metabolites were monitored, a minimum stabilization period was determined to be required
to return endogenous monoamines levels to a stable baseline (Tsai and Chen, 1994; doi: 10.1016/0304-
3940(94)90479-0) (page 5, line 154-156).

Thank you for the point. We agree with you that if the balancing period is too short, this will probably
lead to overestimation of drug brain penetration.

b. The perfusate composition (ACD solution) is unusual, especially for brain microdialysis. Normally,
the perfusate composition tries to mimic the extracellular fluid composition and pH as closely as
possible. ACD solutions are acidic and missing key ions of the extracellular fluid, which will affect
the pH and ion composition of the tissue surrounding the probe. This may compromise the integrity of
the BBB. This is a clear limitation of the study and should be discussed.

Response: Ringer's solution was used as the perfusate for brain dialysis at the beginning. However, a
serious mass matrix effect significantly affected the signal of NHC when Ringer solution was used.

We face some analytical challenges in LC–MS/MS experiments. To resolve this analytical problem,
several perfusates were tried. Finally, ACD solution was selected as the perfusate to replace Ringer
solution to decrease the matrix effect (page 5, line 156-159).

c. The in vitro probe recovery calculation is problematic and may lead to erroneous data manipulation.
This is especially true when two very different tissues (blood and brain) are being compared (AUC
ratios), and because the perfusate composition differs so much from that of the investigated tissues.
The in vivo recoveries should be determined and used for recovery calculations. The authors should
also highlight in the text and tables that the drug concentrations are corrected with recovery values.

Response: The unclear statement has been revised accordingly. Due to the different active dialysis
areas in these two different tissues (blood and brain), the analyte concentrations in blood and brain
were corrected by each correlated in vitro recovery. Usually, calibration of the microdialysis probe is
determined by in vitro relative recovery (RR) (dialysate extraction fraction). Considering the
microenvironment of the position of implantation, the RR in vivo may be different from the RR in
vitro. However, not only the components of extracellular fluid but also the endogenous
biotransformation system may affect recovery. Microdialysis recovery is determined by mass transport
of the solute over the dialysis membrane. In this experiment, the in vitro recoveries of microdialysis
probes in blood and brain were determined and are presented in Supplementary Information Table S4.
(Supplementary Information, Table S4)

95 d. A minor comment: are the stereotaxic coordinates calculated from the top of the skull or dura mater?
Is it the tip of the probe that is extending to -7,5 mm (from dura or skull)? Please specify these details.

Response: The stereotaxic coordinates were calculated from the top of the skull, and the tip of the
probe was inserted into the brain -7.5 mm calculated from the top of the skull. After mounting with a
stereotaxic instrument, the hole was drilled in the skull with a pen-type grinder, and a brain probe was
implanted at the striatum site (+0.2 mm anteroposterior, +3.0 mm mediolateral and -7.5 mm
dorsoventral to bregma) from the skull according to the guide *The Rat Brain in Stereotaxic Coordinates*
(Paxinos, 1982) (page 8, line 285-288).

3) The authors make rather optimistic translational interpretations: they justify the dose of
molnupiravir (100 mg/kg i.v.) by human clinical dose (1600 mg/day p.o.) “multiplied by the
bioavailability”. However, the species differences are not considered in this calculation. Is the dose in
accordance e.g. to the FDA guideline for calculating the human equivalent dose based on the body
surface area of the species? Furthermore, it is important to keep in mind that the microdialysis method
measures the extracellular unbound drug concentrations in a fraction of time (20 min in this case), and
the measured concentrations do not represent exact concentrations at specific time points as often
misinterpreted. Although the authors justify the use of i.v. route, the translational value of the findings

would increase if molnupiravir had been administered perorally as in patients.

Response: The bioavailability of molnupiravir was reported in a study by the European Medicine
Agency (EMA), in which an animal study (mice) was used to evaluate bioavailability. The dosage
regimen molnupiravir (100 mg/kg, i.v.) was transferred from human to rat experiments. ...The dosage
used in this experiment was based on the daily dose used clinically (1600 mg/day/human). The
bioavailability of molnupiravir is 90% (European Medicines Agency, 2022), and the transfer constant
to a rat dosage of molnupiravir was determined (147 mg/kg, i.v.). In the experiment, considering that
molnupiravir is taken once every 12 hours, the dose we used was 100 mg/kg. (page 8, line 267-270)

4) Scientific presentation is incomplete, and some important literature is missing. Some examples:

a. Methods section is presented twice (chapters 3 and 5), why is that?

Response: The Methods section has been revised. We apologize for this mistake due to reformatting
from Nature Communications guidelines to the guidelines of Communications Medicine. (page 8-9)

b. graphical abstract is missing the blood microdialysis

Response: A depiction of the blood microdialysis probe was added in the graphic abstract accordingly.
(graphical abstract)

c. a more thorough description of molnupiravir pharmacology should be given in the Introduction
(molnupiravir – NHC – triphosphated nucleoside analog EIDD-2061)

Response: The unclear statement has been revised accordingly. Molnupiravir is an orally bioavailable
prodrug of the nucleoside analog β -D-N4-hydroxycytidine (synonyms: N4-hydroxycytidine; NHC;
EIDD-1931) that is used to treat SARS-COV-2 infection, and NHC is the active metabolite of
molnupiravir and is widely used in broad-spectrum antiviral drugs. NHC is phosphorylated by host
kinases to activate the intracellular metabolite EIDD-1931-5'-triphosphate (EIDD-2061)(Gordon et al.,
2021; Painter et al., 2019; Painter et al., 2021). The pharmacological mechanism of molnupiravir is
similar to that of remdesivir, which targets the RNA-dependent RNA polymerase (RdRp) enzyme used
by the coronavirus for the transcription and replication of its viral RNA genome (Kabinger et al., 2021).
NHC is inserted into viral RNA to replace uracil, and while RdRp uses NHC-containing RNA as a
template, the enzyme can subsequently incorporate an incorrect nucleotide into the growing RNA
strand, leading to mutagenesis (Zhao et al., 2021). (page 3, line 45-54)

144 d. There are recent reports on the pharmacokinetics of EIDD-1931 and EIDD-2061 that should be
incorporated into the manuscript (e.g., <https://doi.org/10.1016/j.antiviral.2019.104597>; doi:
10.1016/j.ejps.2022.106345). Furthermore, it should be mentioned that molnupiravir is one of the rare
orally administered drugs against COVID-19.

Response: The unclear statement has been revised accordingly. Molnupiravir is a rare orally

bioavailable prodrug of the nucleoside analog β -D-N4-hydroxycytidine (synonyms: N4-
hydroxycytidine; NHC; EIDD-1931) that is used to treat SARS-COV-2 infection, and NHC is the
active metabolite of molnupiravir and is widely used in broad-spectrum antiviral drugs. NHC is
phosphorylated by host kinases to activate the intracellular metabolite EIDD-1931-5'-triphosphate
(EIDD-2061)(Gordon et al., 2021; Painter et al., 2019; Painter et al., 2021). (page 3, lines 44-48) The
articles <https://doi.org/10.1016/j.antiviral.2019.104597> and doi: 10.1016/j.ejps.2022.106345 have
been added to references 3 and 29.

e. row 54: what does “can quick clearance” mean?

Response: The unclear statement has been revised accordingly. Quick clearance means that it can
quickly inhibit virus replication. (page 3, line 57)

f. row 60: effective concentration of NHC against what?

Response: The unclear statement has been revised accordingly. Previous reports showed that the 90%
effective concentration of NHC against SARS-CoV-2 infection in an in vitro study was between 0.5
μ M and 1 μ M (equivalent to 0.13-0.26 μ g/mL) in different cell lines (Sheahan et al., 2020), and the
report showed that the EC₉₀ of NHC was 6 μ M in the Vero 76 cell line (Barnard et al., 2004). (page 3,
line 62-65)

168 g. row 62: “molnupiravir was treated”, how a treatment can be treated?

Response: The unclear statement has been revised accordingly. In an in vivo study, molnupiravir was
administered to animals that were infected with SARS-CoV-2 by oral administration at doses of 128
and 200 mg/kg in ferrets and hamsters, respectively (Abdelnabi et al., 2021; Rosenke et al., 2021). In
addition, this dosage can be maintained at effective concentrations after organ distribution(Douaud et
al., 2022; Krasemann et al., 2022; Pellegrini et al., 2020; Song et al., 2021). (page 3, line 65-67)

175 h. row 65: effective concentration against what and where?

Response: The unclear statement has been revised accordingly. In an in vivo study, animals infected
with SARS-CoV-2 were treated with molnupiravir by oral administration at doses of 128 and 200
178 mg/kg in ferrets and hamsters, respectively (Abdelnabi et al., 2021; Rosenke et al., 2021). In addition,
this dosage inhibits SARS-COV-2(Douaud et al., 2022; Krasemann et al., 2022; Pellegrini et al., 2020;
Song et al., 2021). However, whether the concentration of molnupiravir and NHC is higher than the
effective concentration is still unclear. (page 3, line 65-68)

i. row 73: what does modulation mean here?

Response: The unclear statement has been revised accordingly. The equilibrative nucleoside
transporter (ENT) is a bidirectional transporter(Podgorska et al., 2005), and it mainly transports

nucleosides between the blood and endothelium of the brain (Anderson et al., 1999). A previous report
on computational 3D pharmacophore models and in vitro data suggested that increased uptake of NHC
may occur due to ENT being blocked by NBMPR, resulting in improved efficacy against SARS-CoV-
2 (Miller et al., 2021). However, there is still no in vivo evidence to demonstrate the modulation of the
transporters for molnupiravir and NHC in the BBB. (page 3, line 71-77)

j. row 115: based on the time-concentration curves (fig 2) the compounds seem to follow a two-
compartment model with distinct distribution and elimination phases, thus, please specify which
elimination half-life (alpha, beta) has been calculated

Response: The unclear statement has been revised accordingly. Because molnupiravir was eliminated
quickly and could only be detected in blood for 100 min, if a two-compartment model was used for
calculation, much of the data could not be calculated. WinNonlin Standard Edition software (version
1.1; Scientific Consulting Inc., Apex, NC, USA) was used to calculate the main pharmacokinetic
parameters with a one-compartment model of molnupiravir in blood selected based on Akaike's
Information Criterion (AIC)³⁵ and the noncompartmental model of molnupiravir in the brain and NHC
in the blood and brain. (lines 297-301)

k. rows 120-123: AUC ratio of molnupiravir: NHC is miscalculated ($7228/4214 = 1.71$, not 2.10 as
stated)?

Response: The miscalculation has been revised accordingly. The biotransformation of molnupiravir to
NHC was expressed using $AUC_{NHC}/AUC_{molnupiravir}$ from blood and brain, 1.71 ± 0.08 and $4.50 \pm$
0.22 , respectively. (page 5, line 124-129)

209 l. row 124: it cannot be concluded that the **biotransformation ratio** of molnupiravir to NHC was
210 significantly higher in the brain than in the blood; NHC in the brain is most likely released in the
211 periphery and transported into the brain, not "biotransformed" there.

Response: The word "biodistribution" has been replaced in this statement. The results demonstrated
that the biodistribution ratio in the brain was significantly higher than that in the blood. The
biodistribution ratios (AUC_{brain}/AUC_{blood}) of molnupiravir and NHC transfer into the brain were $0.3 \pm$
0.02% and $0.8 \pm 0.04\%$, respectively (Table 1). These results demonstrated that NHC crossed the
BBB into the brain, and the transfer ratio was between 0.3-0.8%. (page 5, line 128-130)

218 m. rows 126-127: check the numbers (AUC ratio 0.003 and 0.008, i.e., transfer ratios 0.2 and 0,9%?)

Response: The AUC ratio has been used to replace the percentage. The biodistribution ratios
(AUC_{brain}/AUC_{blood}) of molnupiravir and NHC in the brain were $0.3 \pm 0.02\%$ and $0.8 \pm 0.04\%$,
respectively (Table 1). (page 5, line 131-132)

n. row 162: how was the depth of anesthesia controlled?

Response: The rat anesthesia control was determined based on the guidelines of the book “A
GUIDEBOOK FOR CARE AND USE OF LABORATORY ANIMALS” and “GUIDE FOR THE
CARE AND USE OF LABORATORY ANIMALS (8th Edition)”. The toe-pinch reflex was used to
determine the level of anesthesia in rats. (page 8, line 262-264)

o. row 227: how does the IC₅₀ of NHC suggest that NHC penetrated the BBB? Rephrase.

Response: The unclear statement has been revised accordingly. In 2020, Sheahan et al. reported that
NHC was potently antiviral with an average median inhibitory concentration (IC₅₀) of 0.08 μM to 0.3
μM (equivalent to 20.8-78 ng/mL) (Sheahan et al., 2020). Our results suggested that NHC penetrated
the BBB, and the concentration was within or above therapeutic concentrations approximately 300
234 min after molnupiravir administration (100 mg/kg, i.v.). (page 6, line 174-178)

p. rows 230-232: rephrase to clarify the point.

Response: The unclear statement has been revised accordingly. ... with SARS-CoV-2 infection, the
concentrations of NHC in the plasma, saliva, nasal samples, and tear samples were maintained within
or above the 90% effective concentration (EC₉₀) 0.5-1 μM, which is approximately equivalent to 0.13-
0.26 μg/mL of NHC, and the treatment showed protection against SARS-CoV-2 infection (FitzGerald
et al., 2022; Toots et al., 2019). (page 6, line 178-182)

q. row 271: what aspect of molnupiravir and NHC was modulated by the transporter? Please clarify
and rephrase.

Response: The unclear statement has been revised accordingly. In 2023, the study published by Saleh
et al. used LeiCNS-PK3.0, a physiologically based pharmacokinetic (PBPK) model, to predict whether
the concentration of molnupiravir in the brain can reach the effective concentration (Saleh et al., 2023).
They reported that the dosage needs to be 4000 mg twice daily and can be effective in the brain. The
previous data indicated that the EC₅₀ and EC₉₀ values for treating infections with the delta SARS-
COV-2 strain are 1.43 and 4.65 μM (equivalent to 471.9 and 1534 ng/mL), respectively, and the EC₅₀
value for treating infections with the omicron SARS-COV-2 strain is 0.25 μM (equivalent to 82.5
252 ng/mL). In our study, the drug concentration in blood could reach both the EC₅₀ and EC₉₀ for treating
infections with delta and the EC₅₀ for treating infections with omicron; however, the concentration of
NHC in the brain could only reach the EC₉₀ for treating infections with delta for a short time if
administered with molnupiravir 100 mg/kg. (page 7, line 210-219)

r. The authors could better highlight that the IC₅₀ or EC₉₀ values determined in in vitro cellular assays
are usually protein unbound concentrations measured from the cell medium, and these concentrations
are comparable with those measured by in vivo microdialysis (extracellular unbound concentrations).

Response: Our manuscript highlights the IC₅₀ value in an in vitro study. The concentrations of NHC
in both the blood and brain were higher than the IC₅₀ reported in a previous study. In 2020, Sheahan
et al. reported that NHC was had a potent antiviral effect with an average half maximal inhibitory
concentration (IC₅₀) of 0.08 μM to 0.3 μM (equivalent to 20.8-78 ng/mL) (Sheahan et al., 2020). Our
results suggested that NHC penetrated the BBB, and the concentration was within or above therapeutic
concentrations approximately 300 min after molnupiravir administration (100 mg/kg, i.v.). These
results were also consistent with a clinical report that molnupiravir was administered 800 mg twice
daily for patients with SARS-CoV-2 infection, the concentrations of NHC in the plasma, saliva, nasal
samples, and tear samples were maintained within or above the 90% effective concentration (EC₉₀) of
0.5-1 μM, which is approximately equivalent to 0.13-0.26 μg/mL of NHC, and the treatment showed
protection against SARS-CoV-2 infection (FitzGerald et al., 2022; Toots et al., 2019). (page 6, line
174-182)

273 s. discussion on rows 275-280 is purely speculative, not based on the data, and the point is unclear.
Molnupiravir has been shown to be unstable in plasma, thus, it is plausible that most of the NHC found
in the brain is released in the periphery. I recommend removing this paragraph.

Response: The unclear statement has been removed.

278 t. The IC₅₀ values in figure 2 should be removed (because they are not determined in this study). Also,
the figure legends are difficult to interpret (what does “molnupiravir (+molnupiravir)” mean?). Figure
text states that values represent mean ± SEM, but in chapter 3.4. it is said that data have been expressed
as mean ± SD?

Response: The unclear statement has been revised accordingly. The symbols ● and ○ represent
the concentrations of molnupiravir and NHC, respectively, during administered molnupiravir (100
284 mg/kg, i.v.) alone. The symbols ▼ and △ represent the concentrations of molnupiravir and NHC,
respectively, during concomitant administration of molnupiravir (100 mg/kg, i.v.) and NBMPR (15
286 mg/kg, i.v.). Data are expressed as the mean ± standard error of the mean (SEM). (page 15-16, Figure
2)

u. Some of the statistical methods (Mann-Whitney U-test) that are mentioned in the footnote of table
1 are not mentioned in the methods section. Mann-Whitney U-test cannot be combined with Tukey's
post hoc test. Why parametric and nonparametric tests have been mixed, have the authors tested the
normal distribution of the data? The use of different tests should be justified.

Response: The unclear statement has been revised to indicate the use of Student's *t* test with a post hoc
Tukey HSD test to compare the AUC_{brain}/AUC_{blood} of molnupiravir and NHC in the group treated with
molnupiravir (100 mg/kg, i.v.) alone. (page 9, line 309-311; page 16, Table 1)

v. English language needs revision throughout the manuscript

Response: The English language of the manuscript has been revised by American Journal Experts
(verification code ABF5-C0B2-A900-9AB9-669P) for Language Editing Services.

Reviewer #2:

1. Are the authors really investigating the transporter modulation as stated in the title (i.e., can they
prove they are directly modulating the transporters for molnupiravir and NHC)?

Response: Thank you for the comment. The unclear statement has been revised accordingly. To date,
the mechanisms of the transporters of molnupiravir and NHC in the brain are not fully understood. In
this study, we used NBMPR, an ENT inhibitor, to block the transporters of molnupiravir and NHC in
experimental rats. The results showed that the blood-to-brain ratios of molnupiravir and NHC
concentration increased significantly from 0.3% to 0.8% and 0.7% to 1.5%, respectively, in the brain
(Table 1). These results suggested that the bidirectional influx and efflux function of the ENT
transporter was blocked to cause accumulation of molnupiravir and NHC in the brain, confirming a
previous report that molnupiravir and NHC were modulated by nucleoside transporters (CNT and ENT)
(Molina-Arcas et al., 2009; Podgorska et al., 2005; Vlachodimou et al., 2019). When the ENT channel
was blocked, the efflux system was also blocked, and molnupiravir and NHC accumulated in the brain
through CNTs. (page 7, line 225-234)

2. Please see the very recent paper published in European Journal of Pharmaceutical Sciences by
Saleh et al on the prediction of BBB transport and efficacy of Molnupiravir and discuss in the
Discussion: The PBPK LeiCNS-PK3.0 framework predicts Nirmatrelvir (but not Remdesivir or
Molnupiravir) to achieve effective concentrations against SARS-CoV-2 in human brain cells

Response: The unclear statement has been revised, and the reference was added. In 2023, the study
published by Saleh et al. used LeiCNS-PK3.0, a physiologically based pharmacokinetic (PBPK) model,
to predict whether the concentration of molnupiravir in the brain can reach the effective
concentration(Saleh et al., 2023). They reported that the dosage needs to be 4000 mg twice daily and
can be effective in the brain. The previous data indicated that the EC50 and EC90 values for treating
infections with the delta SARS-COV-2 strain are 1.43 and 4.65 μM (equivalent to 471.9 and 1534
326 ng/mL), respectively, and the EC50 value for treating infections with the omicron SARS-COV-2 strain
is 0.25 μM (equivalent to 82.5 ng/mL). In our study, the drug concentration in blood could reach both
the EC50 and EC90 of treating delta infections and the EC50 of treating omicron infections; however,
the NHC in the brain could only reach the EC90 of delta for a short time if administered with
molnupiravir 100 mg/kg. (page 7, line 210-219).

3. Line 136-141: I do not think the authors can identify the biotransformation of molnupiravir to
NHC occurring in the brain using their methods. How can they correct for the amount of NHC

transported across the BBB?

Response: The unclear statement regarding biotransformation has been revised to reflect
biodistribution. The biotransformation and biodistribution of molnupiravir to NHC was expressed
using $AUC_{NHC}/AUC_{molnupiravir}$ from blood and brain, 1.71 ± 0.08 and 4.50 ± 0.22 , respectively. The
results demonstrated that the biodistribution ratio in the brain was significantly higher than that in the
blood. The biodistribution ratios (AUC_{brain}/AUC_{blood}) of molnupiravir and NHC transfer into the brain
were $0.3 \pm 0.02\%$ and $0.8 \pm 0.04\%$, respectively (Table 1). These results demonstrated that NHC
crossed the BBB into the brain, and the transfer ratio was between 0.3-0.8%. (page 5, line 128-134)

4. Line 250-251: Saturability of molnupiravir was not investigated.

Response: The statement regarding the unsaturated phenomenon has been revised. The concentration
of molnupiravir reached T_{max} at 40 min with a maximum concentration of $0.35 \pm 0.03 \mu\text{g/mL}$ and was
rapidly metabolized until undetectable after 80 min. This result indicated a delay in the entry of
molnupiravir into the brain and a rapid decline within the dose regimen. (lines 197-200)

5. Lines 253-254: How is it known NHC is actually crossing the BBB and not being converted from
molnupiravir once present in the brain to NHC?

Response: Since carboxylesterase also exists in the brain, the NHC concentration in the brain may
come from source contributions of biodistribution and biotransformation. To date, the mechanisms of
the transporters of molnupiravir and NHC in the brain are not fully understood. In this study, we used
NBMPR, an ENT inhibitor, to block the transporters of molnupiravir and NHC in experimental rats.
The results showed that the blood-to-brain ratios of molnupiravir and NHC concentrations increased
significantly from 0.3% to 0.8% and 0.7% to 1.5%, respectively, in the brain (Table 1). These results
suggested that the bidirectional influx and efflux function of the ENT transporter was blocked to cause
accumulation of molnupiravir and NHC in the brain, confirming a previous report that molnupiravir
and NHC were modulated by nucleoside transporters (CNT and ENT) (Molina-Arcas et al., 2009;
Podgorska et al., 2005; Vlachodimou et al., 2019). When the ENT channel was blocked, the efflux
system was also blocked, and molnupiravir and NHC accumulated in the brain through CNTs. (page
7, line 225-234)

6. Figure 1: The complete description of the samples in C are not described (i.e., how long after
molnupiravir administration)

Response: The unclear statement has been revised accordingly. (C) Blood dialysate sample containing
NHC (758.5 ng/mL) and molnupiravir (4.6 ng/mL) collected at 140 min after molnupiravir
administration (100 mg/kg, i.v.) (page 14, Figure 1).

7. As ENT inhibition increases drug levels in brain, indicating a role in brain efflux of the drugs,

please discuss or amend the introduction that states ENTs are primarily involved in transport from
blood to the endothelium (and not vice versa) and why the results presented here differ.

Response: The unclear statement has been revised accordingly. These results suggested that the
bidirectional influx and efflux function of the ENT transporter was blocked to cause accumulation of
molnupiravir and NHC in the brain, confirming a previous report that molnupiravir and NHC were
modulated by nucleoside transporters (CNT and ENT) (Molina-Arcas et al., 2009; Podgorska et al.,
2005; Vlachodimou et al., 2019). When the ENT channel was blocked, the efflux system was also
blocked, and molnupiravir and NHC accumulated in the brain through CNTs. (page 7, line 229-234)

Minor Comments

1. Grammatical errors: Lines 77-79, 130-132

Response: The English language of the manuscript has been revised by American Journal Experts
(verification code ABF5-C0B2-A900-9AB9-669P) for Language Editing Services. The microdialysis
probe comprises a semipermeable membrane, and the principle of microdialysis is based on the
concentration gradient between perfusate and its extracellular environment and allows hydrophilic
small molecules to penetrate the membrane. (line 79-81) To investigate the mechanism of the BBB
transporter by which molnupiravir and NHC cross into the brain, an equilibrative nucleoside
transporter inhibitor, NBMPR (10 mg/kg, i.v.) was concomitantly administered with molnupiravir (100
389 mg/kg, i.v.). (lines 137-139)

2. 127, 219, 250, 254: blood-brain barrier has already been defined and abbreviated (BBB)

Response: The abbreviation has been revised accordingly. (line 133, 168, 189, 204 and 206)

3. Is line 206-210 critical for the first sentence of the Discussion?

Response: In this study, the multiple microdialysis technique used in this study was in compliance with
the 3Rs principle (replacement, reduction, and refinement) for experimental animal studies, which
reduced the number of animals needed to collect samples, measure the protein-unbound drug in the
blood vessel and extracellular space of the brain, and continuously monitor the kinetic changes of the
analyte. (lines 151-154)

4. Line 275-280: difficult to follow text; please rephrase

Response: This paragraph was removed to focus the discussion on pharmacokinetics and transporters.

Reviewers' comments:

Reviewer #1 (Remarks to the Author):

The authors have put a lot of work to the revision of their manuscript, and it has been substantially improved by the modifications. However, their answer considering the dysfunctional BBB after the microdialysis probe insertion is insufficient. They claim that neurotransmitter and metabolite levels were stabilized within one hour after probe insertion in their previous studies. This may be true for some transmitters, but extracellular transmitter levels do not prove that the BBB is functional and intact. The authors should take a close look into the seminal microdialysis works, e.g., by M Hammarlund-Udenaes, E de Lange, and M Danhof. An additional experiment with the same experimental setup showing that the BBB function has been restored, e.g. using a low brain-penetrating compound such as atenolol as a reference compound, is needed. The short stabilization period together with the unusual perfusate and in vitro recovery assessment may significantly affect the presented results and lead to erroneous interpretations.

Reviewer #2 (Remarks to the Author):

The authors have addressed all my comments.

Manuscript number: COMMSMED-23-0031-A R2

Transporter modulation of molnupiravir and β -d-N4-hydroxycytidine at the blood–brain barrier in rats

Chun-Hao Chang¹, Wen-Ya Peng¹, Wan-Hsin Lee¹, Ling Yang¹, Tung-Yi Lin¹, Muh-Hwa Yang², Tung-Hu Tsai^{1,3,4*}

Reviewers' comments:

Reviewer #1 (Remarks to the Author):

The authors have put a lot of work to the revision of their manuscript, and it has been substantially improved by the modifications. However, their answer considering the dysfunctional BBB after the microdialysis probe insertion is insufficient. They claim that neurotransmitter and metabolite levels were stabilized within one hour after probe insertion in their previous studies. This may be true for some transmitters, but extracellular transmitter levels do not prove that the BBB is functional and intact. The authors should take a close look into the seminal microdialysis works, e.g., by M Hammarlund-Udenaes, E de Lange, and M Danhof. An additional experiment with the same experimental setup showing that the BBB function has been restored, e.g. using a low brain-penetrating compound such as atenolol as a reference compound, is needed. The short stabilization period together with the unusual perfusate and in vitro recovery assessment may significantly affect the presented results and lead to erroneous interpretations.

Response: Thank you for the comment. When a microdialysis probe is inserted into the brain, there can be a temporary disruption or disturbance of the BBB due to the mechanical trauma caused by the insertion process. This disruption can lead to an increased permeability of the BBB, allowing substances that would normally be restricted from entering the brain to pass through. Microdialysis is a minimally invasive sampling technique and can cause transient dysfunction of the biological barrier. It is widely used to collect unbound protein samples from extracellular tissues or organs in vivo ^{22,23}. (page 3, line 79-81)

However, the BBB has the remarkable ability to rapidly repair itself. Within a few hours after the insertion of a microdialysis probe, the endothelial cells lining the blood vessels begin to repair the damaged areas and restore the integrity of the BBB. This repair

process involves the resealing of tight junctions between the endothelial cells, which effectively re-establishes the selective permeability of the BBB.

The exact molecular and cellular mechanisms underlying the rapid recovery of the BBB are still being investigated. However, it is believed that various factors, including the activation of signaling pathways, recruitment of immune cells, and release of specific molecules, contribute to the repair process.

We agreed with the reviewer's suggestion that the use low brain penetration should be the internal standard to ensure the recovery of BBB function was recovered, however, after searching the literature, we found that many papers also judge whether the blood-brain barrier is restored by monitoring the stability of neurotransmitters in the brain, for example, Xie, R. & Hammarlund-Udenaes, M. Blood-brain barrier equilibration of codeine in rats studied with microdialysis. *Pharmaceutical Research* 15, 570-575 (1998); de Lange et al. Methodological considerations of intracerebral microdialysis in pharmacokinetic studies on drug transport across the blood-brain barrier. *Brain Research Reviews* 25, 27-49 (1997); Yang, L. et al. Modulation of the transport of valproic acid through the blood-brain barrier in rats by the *Gastrodia elata* extracts. *Journal of Ethnopharmacology* 278, 114276 (2021). Therefore, we thought that when neurotransmitters become stable, it suggests that blood-brain barrier function is potentially restored. To monitor neurotransmitters and their metabolites with microdialysis, a minimum one hour stabilization period should be required to balance endogenous monoamine levels back to a stable baseline²⁵. (page 5, line 154-156).

Reviewer #2 (Remarks to the Author):

The authors have addressed all my comments.

Response: Thanks.

REVIEWERS' COMMENTS:

Reviewer #1 (Remarks to the Author):

In their rebuttal letter the authors claim that the blood-brain barrier has a remarkable ability to rapidly repair itself. However, they do not offer any recent references to back up this claim. The papers they are citing to (by Xie and de Lange) do not support the use of transmitter levels as an indicator for BBB integrity; in fact, the old review by de Lange offers several references supporting a long stabilization period (up to 24 hours) after probe insertion. The use of a guide cannula (which was not used in this study) could also shorten the time needed for stabilization.

The authors should take this clear limitation in their study into account in the discussion section - there should be a clear statement that the measured concentrations in the brain ECF may be overestimated because the integrity of the BBB may have been compromised due to the short stabilization period.

Manuscript number: COMMSMED-23-0031-B R3

Transporter modulation of molnupiravir and its metabolite β -D-N4-hydroxycytidine across the blood-brain barrier in a rat

Chun-Hao Chang, Wen-Ya Peng, Wan-Hsin Lee, Ling Yang, Tung-Yi Lin, Muh-Hwa Yang, Tung-Hu Tsai*

Reviewers' comments:

Reviewer #1 (Remarks to the Author):

In their rebuttal letter the authors claim that the blood-brain barrier has a remarkable ability to rapidly repair itself. However, they do not offer any recent references to back up this claim. The papers they are citing to (by Xie and de Lange) do not support the use of transmitter levels as an indicator for BBB integrity; in fact, the old review by de Lange offers several references supporting a long stabilization period (up to 24 hours) after probe insertion. The use of a guide cannula (which was not used in this study) could also shorten the time needed for stabilization.

The authors should take this clear limitation in their study into account in the discussion section - there should be a clear statement that the measured concentrations in the brain ECF may be overestimated because the integrity of the BBB may have been compromised due to the short stabilization period.

Response: Thank you for the comment. We agree with the reviewer's comment that microdialysis has some limitations when the microdialysis probe is inserted into the brain for sampling, which may influence the release of endogenous neurotransmitters from the nervous cell. As a result of the neuro-microenvironment being very sensitive, after drilling a hole in the skull and inserting a microdialysis probe, the blood-brain barrier may be disturbed for a period of time. However, compared to monitoring exogenous chemicals, which are not sensitive, such as neurotransmitter release, a suitable stabilization period is still required. As a result of disturbances in the biological barrier and microenvironment, the analytes may have little overestimated the concentration in the beginning. Although the analytes may be overestimated, the concentration of NHC was greater than the effective concentration more than 10 times. Here, we believe that our research results still support the idea that molnupiravir and NHC can penetrate the BBB into the brain and achieve therapeutic effects. (page 9, line 295-303)